# INRIQ: Implicit Neural Representation for Image Quality Assessment

## Abstract

Measuring the perceptual quality of a degraded signal relative to its pristine reference has long posed a fundamental challenge in signal and image processing. Traditional metrics such as PSNR and SSIM, as well as more recent learning-based approaches like LPIPS and DreamSim, have provided valuable insights, yet each carries inherent shortcomings. Pixel-domain measures like PSNR, for instance, can assign identical scores to two distortions that are visually very different, failing to reflect human perception. Moreover, as the severity of degradation increases or decreases, existing similarity measures often fail to respond proportionally, revealing their limited sensitivity to the true perceptual progression of signal quality. To address these limitations, we introduce **Implicit Neural Representation for Image Quality Assessment (INRIQ)**. The central idea is to first overfit an INR to the reference signal, thereby encoding its structural and frequency content directly in the weight space of the network. We then quantify how the trained INR must adapt in order to approximate the degraded signal, analyzing this process through a Fisher-based sensitivity framework. By shifting the comparison from image or feature space into the parameter space of INRs, our approach captures subtle yet meaningful differences in distortion, offering a more principled and perceptually aligned measure of quality. Our experimental results on entire KADID10k dataset shows that INRIQ is the most sensitive similarity measure for images.

## 1 Introduction

Signals such as images, video, and audio are inherently prone to distortion as they interact with different systems. These distortions may arise from acquisition noise, compression artifacts due to storage limitations, or quality degradation from transmission constraints (Punchihewa & Bailey, 2002). While often unavoidable in practice, such processes can substantially degrade both the fidelity and perceptual quality of the original signal. As a result, measuring the similarity between a degraded signal and its reference has remained a long-standing and fundamental challenge in signal processing. Robust similarity measures are not only critical for assessing system performance, but also play a central role in downstream tasks such as restoration, compression, and enhancement.

In modern machine learning, similarity measures are equally central, shaping both training objectives and evaluation across tasks such as generative modeling, compression, and representation learning. Classical measures like Peak Signal-to-Noise Ratio (PSNR) (Winkler & Mohandas, 2008) and Structural Similarity Index Measure (SSIM) (Wang et al., 2004) remain widely used for their simplicity, but their formulation often misaligns with human perception of signal quality. For instance, two perceptually distinct images may yield the same PSNR, and/or SSIM (See Figure 1), while semantically similar images with slight spatial misalignments may be unfairly penalized. Moreover, these metrics tend to be sensitive only to certain types of degradations, while failing to register others that are perceptually significant. As learning systems increasingly target perceptually rich tasks, the inability of such measures to provide uniformly sensitive and reliable feedback has become a critical bottleneck.

Recent progress has shifted towards *learned perceptual metrics*, such as LPIPS (Zhang et al., 2018), distances induced by large pretrained models like CLIP (Radford et al., 2021), and more recent approaches such as DreamSim (Fu et al., 2023). These methods leverage deep feature spaces to approximate human similarity judgments and have been shown to correlate more strongly with per-

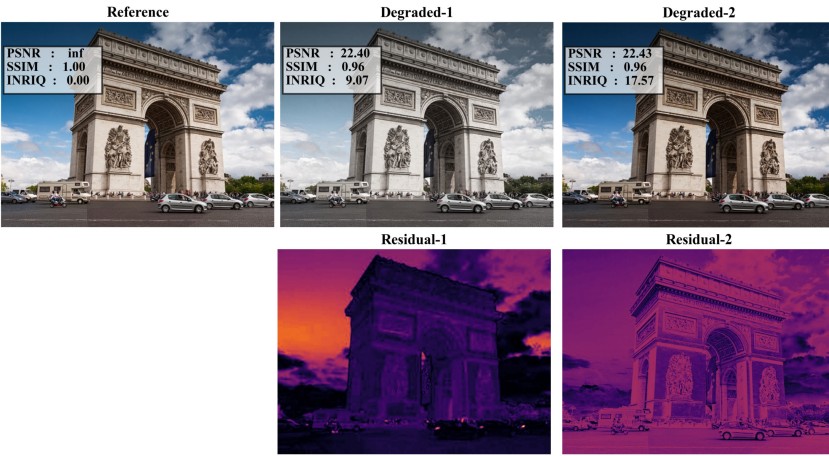

Figure 1: Reference and degraded images with corresponding residual maps for **color saturation** (Degraded-1) and **darkening** (Degraded-2) distortions. Despite yielding nearly identical PSNR and SSIM scores, *INRIQ* clearly separates their perceptual impact. The residual maps illustrate complementary differences: saturation alteration modifies chromatic balance, while darkening affects overall luminance. Conventional metrics overlook these structural and perceptual discrepancies, whereas *INRIQ* captures them effectively.

ceptual scores than classical fidelity measures like PSNR and SSIM. However, they also come with critical limitations. LPIPS is trained on human opinion datasets, CLIP relies on hundreds of millions of image–text pairs, and DreamSim combines multiple foundation models but requires large-scale perceptual annotations for alignment. As a result, these approaches are *highly data-dependent*, inheriting biases from their training corpora that can hinder generalization across domains or to distortions unseen during training. They are also computationally expensive and often operate largely as *black boxes*, with inductive biases tied to their pretraining objectives, making them difficult to analyze theoretically. Moreover, even these perceptual metrics may fail to detect very subtle changes. For instance, an image with only a few noisy or corrupted pixels may appear almost identical to the reference, even to human observers, yet such small distortions can still be important in sensitive domains like compression, restoration, or medical imaging. As these metrics are designed to approximate average human perception, they often overlook these fine-grained structural or frequency-level variations.

This motivates an open question: **can we design a similarity measure that is principled, interpretable, and sensitive to even small structural or frequency-domain changes—while remaining robust and perceptually meaningful, without relying on external training corpora?**

From a signal processing perspective, such a metric should not only account for global fidelity, but also be sensitive to how distortions alter structural and frequency content, as they often appear as high-frequency losses (e.g., blurring) or low-frequency shifts (e.g., banding and blocking). An effective measure must therefore bridge pixel-level fidelity and perceptual alignment in a unified framework, while retaining interpretability.

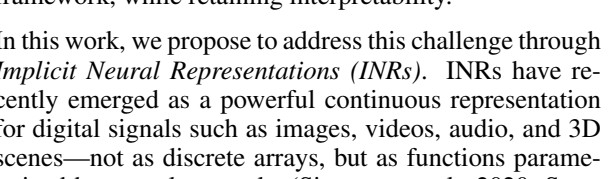
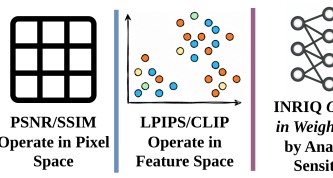

Figure 2: Compared to the pixel space, and feature space measures, INRIQ operates in weight space by analyzing sensitivity.

In this work, we propose to address this challenge through *Implicit Neural Representations (INRs)*. INRs have recently emerged as a powerful continuous representation for digital signals such as images, videos, audio, and 3D scenes—not as discrete arrays, but as functions parameterized by neural networks (Sitzmann et al., 2020; Saragadam et al., 2023; Ramasinghe & Lucey, 2022). By overfitting an INR to a signal, one obtains a compact representation whose parameters implicitly encode both global structure and fine detail. Crucially, INRs exhibit a well-documented *spectral bias*: they recover low-frequency components of a signal early in training, while high-frequency details emerge later (Rahaman et al., 2019). This

progression resonates strongly with human perception, which also prioritizes coarse structure before attending to fine texture. Additionally, Figure 2 presents a summary of the distinct types of similarity measures based on how they operate.

Building on this insight, we introduce a new similarity framework in which perceptual quality is characterized by the trajectory of the INR's *weight space*. Our key idea is simple: once an INR has been fitted to a reference signal, we examine the parameter changes required for it to approximate a degraded version of that signal, measured with respect to the INR parameters of the reference. We quantify this adaptation through the *Fisher information matrix (Fis, 2015)*, which precisely captures the sensitivity of the INR's parameters to the degraded signal. In doing so, we transform the quality assessment problem from comparing pixels or pre-trained features into comparing how two signals are *encoded and optimized* within the same universal approximator.

Unlike conventional perceptual metrics that may be selectively sensitive to certain distortions, our Fisher-based INR metric adapts to the signal itself, offering a principled measure that reflects both structural and frequency-level differences. Moreover, it requires no external supervision or large-scale training, but instead leverages the intrinsic inductive biases of INRs. Conceptually, this reframes similarity evaluation as a problem of analyzing the *sensitivity dynamics of a learned representation*, which naturally aligns with perceptual distinctions that conventional metrics fail to capture.

In summary, this work reframes the evaluation of similarity through the inductive biases of implicit neural representations. By measuring *how signals are represented rather than how they appear in pixels*, we bridge the gap between principled signal processing and perceptually aligned evaluation, introducing a robust and general-purpose family of metrics for machine learning.

## 2 RELATED WORKS

**Classical signal fidelity metrics.** Early approaches to measuring similarity between signals relied primarily on pixelwise fidelity measures. PSNR and Mean Squared Error (MSE) remain widely used due to their mathematical simplicity, but they fail to capture perceptual similarity and often reward visually implausible reconstructions (Wang et al., 2004). The Structural Similarity Measure (SSIM) introduced luminance, contrast, and structural components, offering improved correlation with human perception (Wang et al., 2004). Extensions such as MS-SSIM (Wang et al., 2003) and FSIM (Zhang et al., 2011) further refined these ideas. Nevertheless, these measures are fundamentally hand-crafted and limited in their ability to handle complex perceptual phenomena, especially in tasks involving semantic changes or geometric misalignment.

**Learned perceptual metrics.** Recent work has shifted towards data-driven approaches that leverage deep neural networks as feature extractors. The Learned Perceptual Image Patch Similarity (LPIPS) metric (Zhang et al., 2018) computes distances in feature spaces of pretrained vision models and shows strong alignment with human judgments. Other metrics, such as DISTS (Ding et al., 2020) and PieAPP (Prashnani et al., 2018), explicitly train networks to predict perceptual similarity scores. More recent approaches build on foundation and multimodal models: CLIP-based similarities (Radford et al., 2021), and fine-tuned variants thereof (e.g. adversarially robust R-CLIP) have been used to induce more stable perceptual distances (Croce et al., 2025). Beyond CLIP, Lip-Sim introduces a 1-Lipschitz backbone to provide certified robustness in similarity computations (Ghazanfari et al., 2023), while GLIPS fuses global and local similarity measures with attention and distributional alignment to evaluate AI-generated images more faithfully (Aziz et al., 2024). In imaging settings with scarce annotated data, ConIQA uses consistency training and semi-supervised learning to achieve strong IQA alignment with fewer labels (Eybposh et al., 2024).

**Implicit Neural Representations.** Parallel to these developments, INRs have emerged as powerful tools for encoding signals as continuous functions parameterized by neural networks (Sitzmann et al., 2020; Mildenhall et al., 2021). INRs exhibit a well-documented spectral bias (Rahaman et al., 2019), which aligns naturally with human perceptual sensitivity to low- versus high-frequency components. They have been successfully applied in areas such as image compression (Dupont et al., 2021; Zhang et al., 2023; Guo et al., 2023; Rezasoltani & Qureshi, 2024), generative modeling (Park et al., 2024; Bandyopadhyay et al., 2024), and view synthesis (Mildenhall et al., 2021; Chen et al., 2024). Additionally, a few works explore INR latent codes for downstream tasks (Xu et al., 2022;

Ashkenazi & Treister, 2024); however, a systematic use of INR representation geometry—weights, activations, or derivative-based structures—for defining general-purpose similarity metrics remains unexplored. Our work builds on this gap, introducing INR-based similarity metric as a principled and domain-agnostic a more degradation sensitive alternative to both classical and learned perceptual approaches.

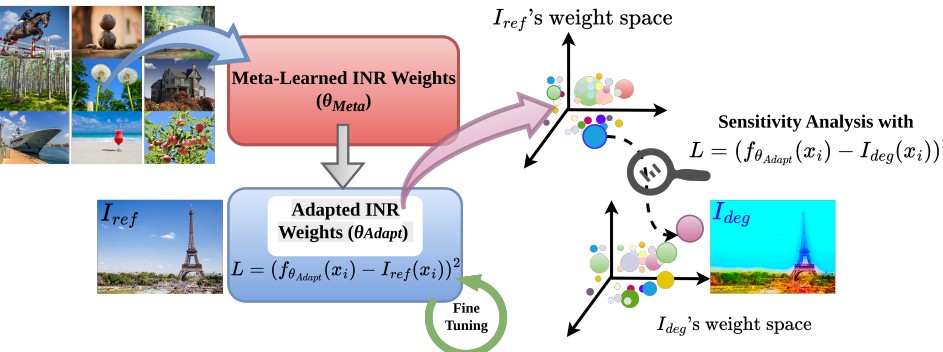

Figure 3: **Operational Mechanism of INRIQ:** INRIQ begins with meta-learned INR weights trained on the Vimeo90k dataset. Given a reference–degraded pair, the INR is first fine-tuned on the reference image $I_{\text{ref}}$ until overfitting. Then, assuming the degraded image $I_{\text{deg}}$ as the target, the method analyzes the required weight perturbations to approximate $I_{\text{deg}}$. Finally, Fisher information and symmetric KL divergence are employed to quantify the INRIQ score.

## 3 METHODOLOGY

### 3.1 LEVERAGING INRs FOR MEASURING SIGNAL SIMILARITY

When leveraging INRs for signal quality assessment, the central idea is to use the INR itself as the proxy for the signal. An INR does not store the signal directly as discrete samples; instead, it learns a continuous mapping parameterized by network weights that reproduce the signal. Training an INR on two different images therefore results in two distinct sets of parameters, each optimized to capture the unique structure of its corresponding signal. If we then define a function that maps a pair of such parameter spaces to a real number, this function can serve as a similarity (or dissimilarity) measure between the underlying signals. In this way, the problem of signal quality assessment is transformed into one of comparing the learned parameterizations of their corresponding INRs.

Now, consider the case where the first signal is a reference and the second is its degraded version. The resulting similarity score should ideally capture how much the INR trained on the degraded signal diverges from that of the reference. A critical question then arises: *which aspect of the INR should serve as the basis for comparison?* Possible candidates include the learned weights, biases, or activations at various layers; alternatively, one might compare combinations of parameters or even the outputs of the networks themselves. The choice of representation is fundamental, as it dictates both the discriminative power of the metric and its sensitivity to different types of degradations.

In our initial experiments, we observed that applying direct distance measures (e.g., $L_2$, cosine) between weight and bias spaces did not yield a perceptually meaningful or reliable quality score when tested on samples of the KADID-10k (Lin et al., 2019) dataset. This is likely because parameters implicitly reflect optimization trajectories rather than perceptual differences, and can therefore diverge without capturing degradation severity. Motivated by prior work on neural network similarity (Klabunde et al., 2025), we adopt layer-wise comparisons using activations, i.e., a notion of *representational similarity*. However, measures such as Canonical Correlation Analysis (CCA) (Pillow Lab, 2019), Centered Kernel Alignment (CKA) (Williams, 2024), and their variants (Klabunde et al., 2025) did not produce a measure that is consistently correlated with the levels of distortion in KADID-10k, as these correlation-based metrics capture broad representational alignment and may be insensitive to the fine-grained spectral degradations introduced by distortions.

## 3.2 GRADIENT-SENSITIVITY VIEW OF INRS

The shortcomings of weight- and activation-based comparisons suggest that what matters is not the static representation of the INR, but how the network is shaped by the signal it is trained on. This led us to a simple yet powerful idea: instead of comparing what the INR has already learned, we compare how it *tries to learn*. During training, the gradients of the loss dictate the directions in which the parameters would move in order to better fit the signal. These gradients are not arbitrary—they are directly determined by the content of the signal itself. For example, once an INR has been well trained on a reference image, it can reproduce the reference RGB value at any input coordinate regardless of the resolution. If we now replace the reference target with a degraded version of the same image and compute a single backpropagation step, the resulting gradients reveal how the weights and biases of the reference-trained INR would have to change to approximate the degraded signal.

**Fisher Diagonal and KL Divergence.** Our metric is constructed by examining how the parameter sensitivities of an INR change when evaluated on degraded images, relative to a reference image. Let $f_\theta : \Omega \subset \mathbb{R}^d \rightarrow \mathbb{R}^c$ denote an INR with parameters $\theta \in \mathbb{R}^P$, mapping a coordinate $u \in \Omega$ (e.g., pixel location) to an output signal value in $\mathbb{R}^c$ (e.g., RGB intensities). We first *fully train* this network on a single reference signal $x^{\mathrm{ref}} : \Omega \rightarrow \mathbb{R}^c$, yielding parameters $\theta^{\mathrm{ref}}$. This reference INR is then frozen and serves as the anchor model for all comparisons.

**Per-pixel gradient computation.** Given another signal $x : \Omega \rightarrow \mathbb{R}^c$ (which may be the reference itself or a degraded variant), we evaluate the frozen reference INR and measure the squared error at each pixel: $\ell(u; \theta^{\mathrm{ref}}) = \frac{1}{2} \left\| f_{\theta^{\mathrm{ref}}}(u) - x(u) \right\|_2^2$. For each coordinate $u$, we compute the gradient of this loss with respect to the network parameters: $g(u; \theta^{\mathrm{ref}}) = \nabla_\theta \ell(u; \theta^{\mathrm{ref}}) \in \mathbb{R}^P$. Therefore, every pixel provides a $P$-dimensional gradient vector describing how the frozen INR's parameters would need to change in order to better match that pixel.

**Diagonal Fisher information.** The Fisher information matrix (Fu et al., 2023; Ghojogh et al., 2019; Li et al., 2025) is classically defined as the covariance of these gradients over the data distribution. In our setting, this corresponds to aggregating per-pixel gradients across the full coordinate grid. For computational tractability, instead of forming the full $P \times P$ Fisher matrix, we retain only to the diagonal entries as an approximation to the exact Fisher matric, which is computed as $F_{\mathrm{diag}}(x) = \frac{1}{|\Omega|} \sum_{u \in \Omega} g(u; \theta^{\mathrm{ref}}) \odot g(u; \theta^{\mathrm{ref}})$ where $\odot$ denotes elementwise multiplication. The result is a vector in $\mathbb{R}^P$, whose elements are non-negative, and each component $F_{\mathrm{diag},j}(x)$ reflects the average squared sensitivity of parameter $j$ when the INR processes image $x$. Intuitively, this captures which parameters are "important" for explaining the content of a given image under the reference INR.

**Approximate KL divergence between Fisher diagonals.** Having computed the Fisher diagonal for the reference image, $F_{\mathrm{diag}}^{\mathrm{ref}} = F_{\mathrm{diag}}(x^{\mathrm{ref}})$ and for a degraded image, $F_{\mathrm{diag}}^{\mathrm{deg}} = F_{\mathrm{diag}}(x^{\mathrm{deg}})$ we would like to measure how different these sensitivity profiles are. A common perspective in information geometry is to view the Fisher information as a local curvature or precision matrix of an underlying Gaussian approximation to the loss landscape (Fis, 2015). Motivated by this, we *approximate* each Fisher diagonal as defining the precision of a diagonal Gaussian, and use the corresponding KL divergence (Jain, 2018; Kurt, 2017) as a practical discrepancy measure: $\mathrm{KL}\left(F^{\mathrm{deg}} \| F^{\mathrm{ref}}\right) \approx \frac{1}{2P} \sum_{j=1}^P \left( \frac{F_{\mathrm{diag},j}^{\mathrm{ref}}}{F_{\mathrm{diag},j}^{\mathrm{deg}}} - 1 - \log \frac{F_{\mathrm{diag},j}^{\mathrm{ref}}}{F_{\mathrm{diag},j}^{\mathrm{deg}}} \right)$. As KL divergence is not symmetric in its arguments, a direct use would violate the desirable property $\mathrm{metric}(a, b) = \mathrm{metric}(b, a)$. To enforce this, we define our similarity measure, which we call **INRIQ**, as the symmetrized form: $\mathrm{INRIQ}\left(F^{\mathrm{ref}}, F^{\mathrm{deg}}\right) = \frac{1}{2}\left( \mathrm{KL}\left(F^{\mathrm{ref}} \| F^{\mathrm{deg}}\right) + \mathrm{KL}\left(F^{\mathrm{deg}} \| F^{\mathrm{ref}}\right) \right)$. The overall process is illustrated in Figure 3. This symmetric divergence serves as a proxy distance between Fisher diagonals. Although based on a Gaussian approximation, it provides a stable, interpretable, and symmetric measure of how the frozen INR's parameter sensitivities differ when processing reference versus degraded images.

**Interpretation.** This procedure can be viewed as follows: we first identify how the reference INR distributes its parameter sensitivities when fitting the clean signal. We then ask: *if we expose the same frozen INR to a degraded image, do the sensitivities remain similar, or do they change drastically?* If the degraded image is perceptually close to the reference, then its Fisher diagonal will

closely match $F_{\text{diag}}^{\text{ref}}$, resulting in a small KL divergence. If distortions significantly alter the structure of the signal, the induced Fisher diagonal will differ, and the KL divergence will be large. In this way, the Fisher–KL metric provides a parameter-space analogue of perceptual similarity, grounded in the information geometry of INRs. Unlike pixelwise measures such as PSNR or SSIM, which only compare intensities directly, our metric evaluates how the underlying *representation* perceives differences through its parameter sensitivities.

### 3.3 META-LEARNING FOR ENHANCED ADAPTATION

Training an INR typically begins from scratch, with network weights randomly initialized and subsequently optimized through backpropagation to fit the target signal. While this procedure is straightforward, it often requires many iterations before the network converges, as random initialization provides no prior knowledge of signal structure. Meta-learning offers a remedy by aiming to learn a good initialization that can quickly adapt to new signals. Instead of starting from arbitrary weights, the network is initialized from parameters that have already been shaped by exposure to a wide distribution of signals. Prior work (Strümpler et al., 2022; Dupont et al., 2022; Lee et al., 2021) has shown that such meta-learned initializations significantly accelerate INR training. Following this approach, we adopt the strategy of (Strümpler et al., 2022) and train our INR with a meta-learning framework on the Vimeo90k dataset (Xue et al., 2019). This provides an initialization that captures general signal statistics and allows faster and more reliable adaptation when fitting to new signals.

### 3.4 RELATIVE SENSITIVITY ANALYSIS

The goal of introducing an INR-based similarity measure, derived through sensitivity analysis, is to enhance the detection of degradations that existing metrics often fail to capture with sufficient precision. Instead of relying solely on pixel- or feature-space differences, the INR framework quantifies the extent to which the network's parameters must shift from their reference configuration in order to represent a degraded signal. This shift, characterized through the diagonal Fisher information and the symmetric KL divergence of parameter gradients, directly encodes the sensitivity of the representation, thereby enabling improved discrimination of subtle or structurally complex degradations.

To study how each similarity measure reflects incremental increases in distortion severity, we introduce a *relative sensitivity analysis*. The intuition is that a similarity measure should ideally exhibit stronger response as the change of level of distortion becomes more severe.

Formally, let $M$ denotes a similarity measure (e.g., PSNR, SSIM, LPIPS), and let $d_i$ denote the $i$-th distortion level for a given distortion type, with $i = 1, \ldots, L$ where $L$ is the number of levels (five in KADID 10k). The *relative sensitivity between two consecutive levels* is defined as:

$$S(M, i) = \left| \frac{\Delta M / M_i}{\Delta d / d_i} \right| = \left| \frac{\frac{M_{i+1} - M_i}{M_i}}{\frac{d_{i+1} - d_i}{d_i}} \right|, \tag{1}$$

where $M_i$ is the value of $M$ at distortion level $d_i$.

The interpretation of this formulation is as follows:

- The numerator $\frac{M_{i+1} - M_i}{M_i}$ measures the *relative change in the similarity measure* between two successive distortion levels.

- The denominator $\frac{d_{i+1} - d_i}{d_i}$ measures the *relative change in distortion severity*.

- The ratio $S(M, i)$ captures how much the similarity measure changes relative to its current value when the distortion level increases relative to its current severity. In other words, it expresses the *relative responsiveness of the measure to a relative increase in distortion*. Since some similarity measures are designed to decrease with distortion (e.g., PSNR, SSIM) while others may increase (e.g., LPIPS, error-type measures), the raw value of $S(M, i)$ can be either positive or negative. To ensure consistency across different measures, we take the absolute value, so that sensitivity always reflects the magnitude of responsiveness, independent of direction.

To summarize across all levels, the *average sensitivity* is defined as:

$$\bar{S}(M) = \frac{1}{L-1} \sum_{i=1}^{L-1} S(M, i).$$

(2)

This aggregated value $\bar{S}(M)$ provides a single indicator of how consistently a similarity measure responds to increasing distortion. In order to ensure numerical stability, cases where $M_i = 0$ or where values become infinite (e.g., PSNR for identical images) are excluded from the averaging. In summary, relative sensitivity analysis provides a principled way to evaluate similarity measures by quantifying how strongly they react to progressive degradations. A desirable similarity measure will yield higher sensitivity values for larger degradations, while still reflecting subtle but perceptible differences at lower levels.

## 4 RESULTS

For experiments, we utilized the KADID-10K dataset, which provides 81 unique images with each having 25 types of degradations and each degradation contains five levels of severity. We first utilized (Strümpler et al., 2022)'s approach to learn meta-learned weights. Then, each reference image is overfitted using the meta-learned weights as the initialization. Once the INR for the reference signal has been obtained, we compute how the parameters must shift in order to represent its distorted counterparts. These updates are then analyzed through the diagonal Fisher information matrix, and the similarity between reference and degraded signals is quantified using the symmetric KL divergence between their corresponding parameter gradient distributions. This procedure yields our proposed INR-based similarity measure. Then, as outlined in Equation (1), and Equation (2), sensitivies have been analyzed against conventional measures such as PSNR, SSIM, MS-SSIM, LPIPS, and CLIP across all distortion types and severity levels in KADID-10K. Among the analyzed degradation types, average results for 14 types of degradations are shown in Table 1. As can be seen *INRIQ* consistently obtains higher sensitivity compared to existing measures.

Table 1: Average relative sensitivity across KADID 10k for selected distortions. Higher value indicates better sensitivity. Best per distortion is highlighted.

| Distortion | PSNR | SSIM | LPIPS | MS-SSIM | CLIP | *INRIQ* |
|---|---|---|---|---|---|---|
| Lens blur | 0.1365 | 0.2210 | 0.9040 | 0.1152 | 0.0641 | **1.4446** |
| Motion blur | 0.1501 | 0.1101 | 1.6880 | 0.0348 | 0.0261 | **2.6382** |
| Color shift | 0.0463 | 0.0090 | **0.4466** | 0.0131 | 0.0239 | 0.4105 |
| Color quant. | 0.1998 | 0.2068 | 0.9754 | 0.0466 | 0.0367 | **1.9703** |
| JPEG2000 | 0.1103 | 0.0811 | 1.3958 | 0.0310 | 0.0312 | **3.5931** |
| JPEG | 0.1737 | 0.1775 | **1.7675** | 0.0962 | 0.1157 | 1.6321 |
| White noise | 0.1187 | 0.2053 | 0.5515 | 0.0525 | 0.0124 | **1.0649** |
| Impulse noise | 0.1105 | 0.1545 | 0.5148 | 0.0508 | 0.0128 | **1.1875** |
| Brighten | 0.3235 | 0.1805 | 2.2387 | 0.1014 | 0.0347 | **2.6364** |
| Darken | 0.2485 | 0.1918 | 2.7168 | 0.0696 | 0.0205 | **3.2511** |
| Jitter | 0.1014 | 0.1052 | 2.1867 | 0.0326 | 0.0420 | **1.4881** |
| Pixelate | 0.0356 | 0.0470 | 0.3815 | 0.0187 | 0.0130 | **0.2285** |
| High sharpen | 0.1936 | 0.2151 | 0.6720 | 0.1217 | 0.0478 | **1.2232** |
| Contrast change | 0.3142 | 0.1863 | 2.3827 | 0.0685 | 0.0214 | **2.7573** |

As shown in Table 1, *INRIQ* achieves the strongest sensitivity for **blur distortions**, especially motion blur, where it surpasses all baselines. For **color distortions**, it captures quantization artifacts that conventional metrics largely miss. In **compression artifacts**, *INRIQ* substantially outperforms others for JPEG2000, demonstrating robustness to structural degradations. It also provides clear advantages for **brightness changes** and **contrast**, while maintaining strong responses to **noise** and **spatial distortions**. Overall, *INRIQ* offers consistent improvements across distortion types, where existing measures remain flat or fail to reflect severity. A sample image from the KADID-10k

dataset for JPEG2000 compression is shown in Figure 4. The corresponding compression ratios are displayed at the top of each figure, except for the reference. Since the INR encodes the reference image, even a single pixel change in pixel space requires the INR weights to shift, thereby capturing even subtle differences. For instance, a perceptual gap becomes clearly visible when transitioning from Level 4 to Level 5; however, PSNR's drop may fail to capture this change, while INRIQ increases from 12.09 to 23.97 as the image quality worsens. A similar trend can also be observed with LPIPS.

Additionally, a color diffusion example is shown in Figure 5. A clear perceptual transition can be observed from Level 1 to Level 2, where **INRIQ** increases from 0.34 to 9.68, with a relative sensitivity of 13.67. This is because many pixel colors have changed, requiring substantial movement in the reference INR's weights.

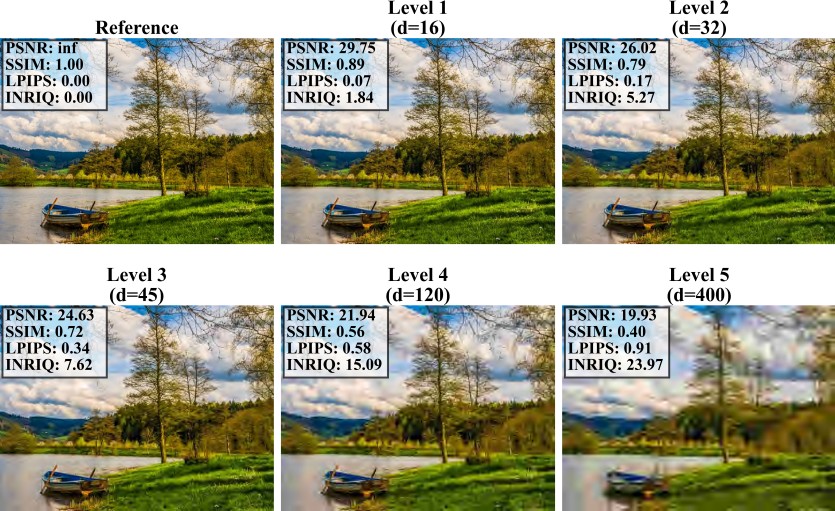

Figure 4: Progression of **JPEG2000 compression** across five severity levels in KADID 10k. IN-RIQ shows the strongest transition sensitivities, notably at **Level 1→2** (**1.86** vs. LPIPS 1.41, PSNR 0.13, SSIM 0.11) and **Level 3→4** (**0.59** vs. LPIPS 0.42, PSNR 0.07, SSIM 0.13). These results demonstrate INRIQ's ability to capture both subtle and mid-level degradations more effectively than conventional metrics.

**INRIQ** can also be used to measure non-standard distortions such as artifacts arising from under-sampling of Magnetic Resonance Images (MRI) (Heckel et al., 2024). We illustrate this on a couple of samples from the FastMRI (Zbontar et al., 2018) dataset by applying under-sampling factors of $2\times$ and $8\times$. The average sensitivities of PSNR, SSIM, LPIPS and **INRIQ**, calculated from the first row of Fig. 6 are 0.0414, 0.0403 0.438, and 0.616, respectively. This shows **INRIQ** is more sensitive than competing metrics to the distortions produced by under-sampling MRI images.

## 5 CONCLUSION

We introduced **INRIQ**, a Fisher-based similarity measure that leverages implicit neural representations (INRs) to evaluate perceptual quality. Instead of comparing pixels or pretrained features, INRIQ quantifies how the parameters of an INR trained on a reference signal must adapt to approximate its degraded counterpart. This reframing allows sensitivity to both structural and frequency-level distortions, while remaining interpretable and symmetric by design. Our experiments on the KADID-10K dataset demonstrate that INRIQ consistently surpasses classical and learned metrics such as PSNR, SSIM, MS-SSIM, LPIPS, and CLIP across a wide variety of distortions. In particular, it shows superior responsiveness to progressive degradation, capturing subtle variations in blur, color shifts, compression artifacts, and brightness changes that other measures often fail to detect. Beyond outperforming existing baselines, INRIQ requires no large-scale supervision or external pre-trained models, relying instead on the intrinsic inductive biases of INRs. This makes it lightweight, domain-agnostic, and theoretically grounded. We believe these findings establish representation-driven sensitivity analysis as a powerful foundation for future quality metrics, with potential extensions to higher-order Fisher structures, video signals, and downstream perceptual optimization tasks.

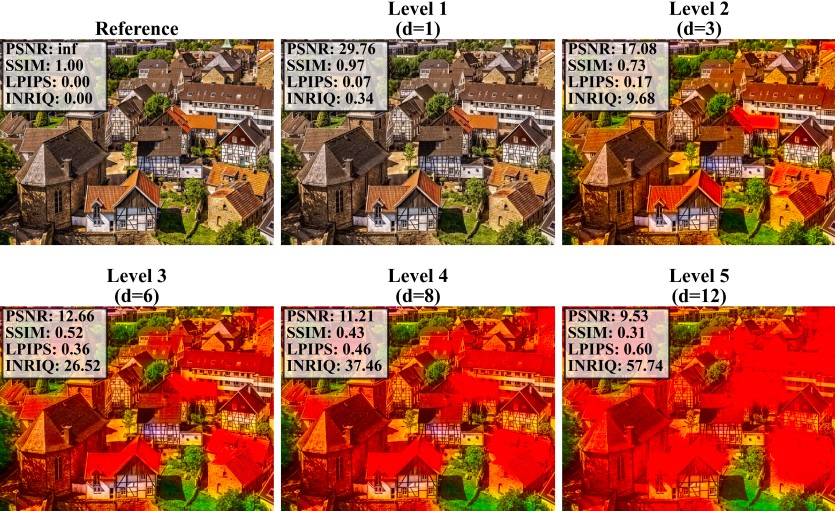

Figure 5: Progression of **Color diffusion** distortion across five severity levels in KADID 10k. IN-RIQ shows exceptional sensitivity at **Level 1→2** (**13.67** vs. LPIPS 0.76, PSNR 0.21, SSIM 0.12), capturing a drastic perceptual change that conventional metrics severely underestimate. At higher degradations, INRIQ remains more responsive, e.g., **Level 3→4** (**1.24** vs. LPIPS 0.79, PSNR 0.34, SSIM 0.53). This demonstrates INRIQ's ability to detect both subtle and severe degradations in color diffusion, outperforming pixel- and feature-based baselines.

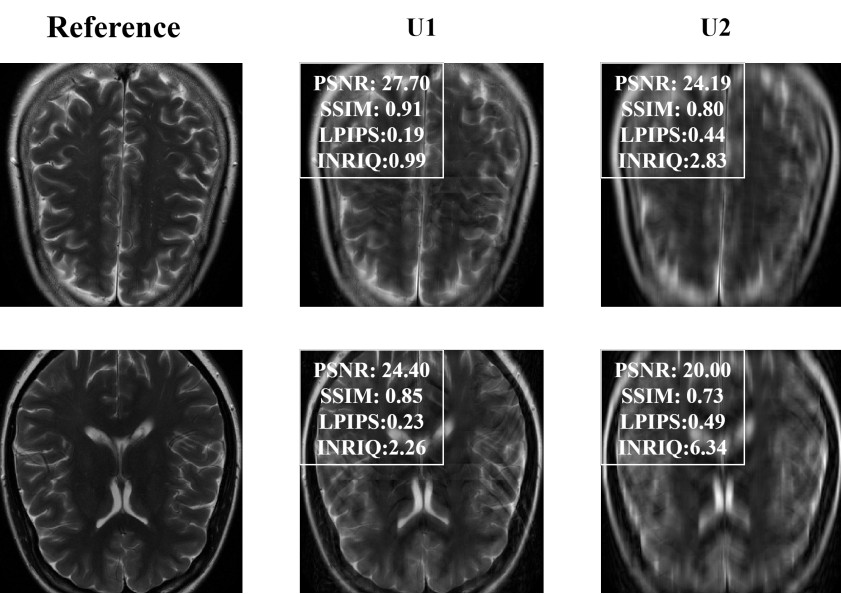

Figure 6: Progression of **Under-sampling** distortion in MRI images. U1 refers to an under-sampling factor of $2\times$ while U2 refers to a factor of $8\times$. INRIQ is more senstive to the MRI distortions than the other metrics.

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

# A APPENDIX

## A.1 EXAMPLE CASES

In the following example cases from KADID 10K dataset, the sensitivities are obtained through Equation (1) and Equation (2).

### A.1.1 MULTIPLICATIVE NOISE (DISTORTION 14)

Multiplicative noise severely corrupts images by introducing signal-dependent speckle patterns that are difficult to detect with conventional quality measures. Across all transitions, our metric INRIQ exhibits significantly higher sensitivity compared to PSNR, SSIM, LPIPS, MS-SSIM, and CLIP. The transitions are shown in Figure 7

In particular, at the early degradation stage (**Level 1→2**), INRIQ attains a sensitivity of **1.44**, while LPIPS remains at 0.28 and PSNR/SSIM are below 0.05, revealing a clear ability to detect subtle multiplicative noise. As severity increases, INRIQ continues to dominate: at **Level 2→3**, it reaches **1.06**, and in the mid-to-late transitions (**Level 3→4** and **Level 4→5**), INRIQ yields **0.99** and **0.93**, respectively, substantially exceeding baselines that remain below 0.30. These results confirm that INRIQ robustly captures both subtle and progressive changes caused by multiplicative noise, where conventional pixel- and feature-space metrics underestimate the perceptual impact.

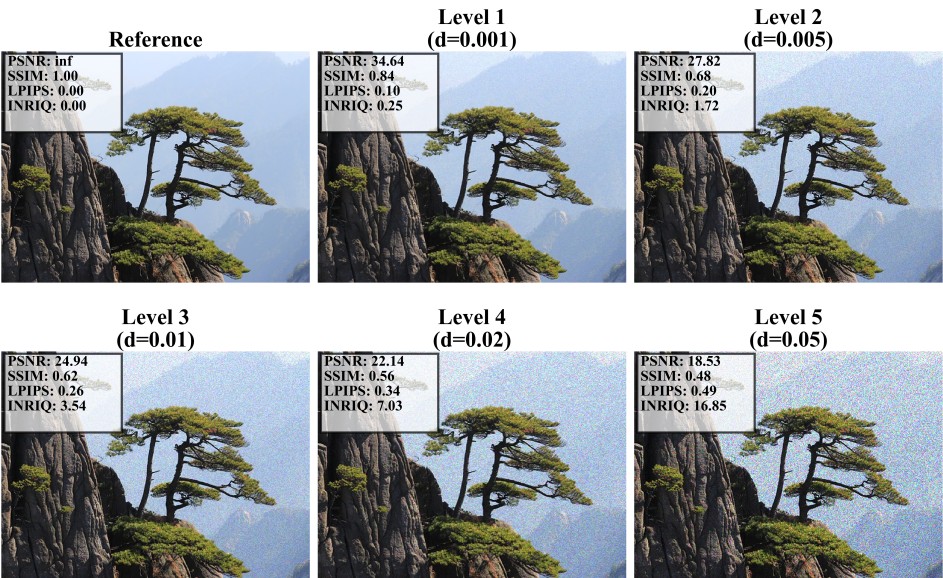

Figure 7: Visual progression of **Multiplicative noise** across five severity levels. INRIQ shows markedly higher sensitivity than conventional metrics, especially in early transitions where distortions are subtle but perceptually evident.

### A.1.2 IMPULSE NOISE

Impulse noise introduces salt-and-pepper style pixel corruption, which is particularly challenging for pixel-based similarity measures. As shown in Figure 8, INRIQ provides consistently higher sensitivity across all severity transitions compared to PSNR, SSIM, LPIPS, MS-SSIM, and CLIP.

At the early stage (**Level 1→2**), INRIQ achieves a sensitivity of **1.58**, whereas LPIPS only reaches 0.57 and PSNR/SSIM remain below 0.05. This demonstrates that INRIQ is highly effective in detecting subtle salt-and-pepper artifacts that are largely underestimated by traditional metrics. As distortion severity grows, INRIQ continues to dominate: for **Level 2→3**, it records **1.28**, and for later transitions (**Level 3→4** and **Level 4→5**), it still maintains **1.08** and **1.03**, while competing metrics remain significantly lower. Overall, INRIQ consistently captures the perceptual progression of impulse noise more robustly than pixel- or feature-based baselines.

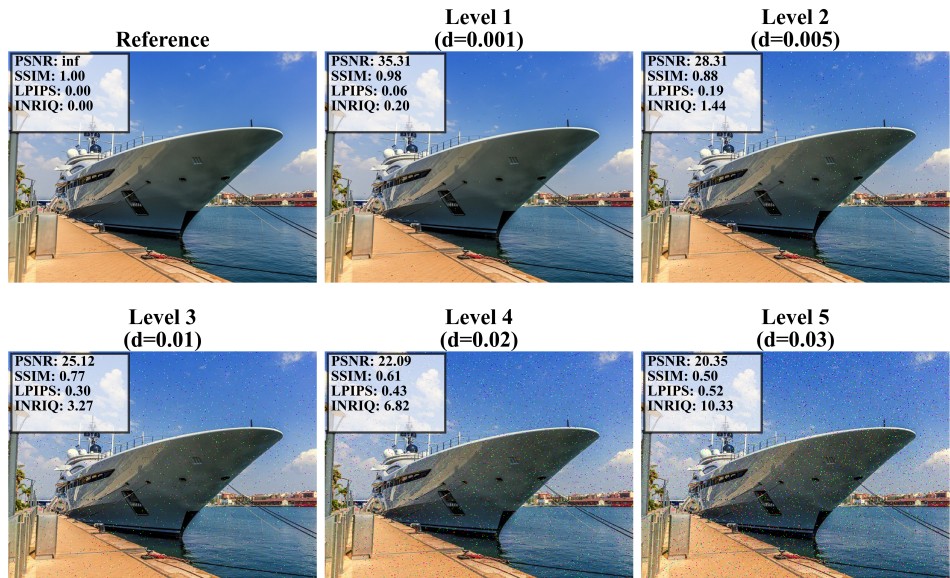

Figure 8: Visual progression of **Impulse noise** across five severity levels. INRIQ demonstrates substantially higher sensitivity, particularly in the early stages (Level 1→2), where conventional metrics largely underestimate the perceptual impact.

### A.1.3 WHITE NOISE

White Gaussian noise directly perturbs pixel intensities, often producing subtle degradations at low severity that are difficult for classical metrics to capture. As illustrated in Figure 9, INRIQ demonstrates consistently higher sensitivity across all distortion transitions compared to PSNR, SSIM, LPIPS, MS-SSIM, and CLIP.

At the early stage (**Level 1→2**), INRIQ attains a sensitivity of **1.18**, whereas LPIPS is limited to 0.49 and PSNR/SSIM remain below 0.20. This shows INRIQ's strong responsiveness to subtle noise perturbations that conventional metrics largely underestimate. As degradation increases, INRIQ continues to dominate: for **Level 2→3**, it reaches **1.08**, and for later stages (**Level 3→4** and **Level 4→5**), it still yields **1.03** and **0.95**, whereas baselines remain significantly smaller. Overall, INRIQ effectively tracks both early and progressive changes under white noise, aligning with perceptual degradation more reliably than standard measures.

### A.1.4 MOTION BLUR

Motion blur introduces directional smearing that significantly alters image structure while maintaining global smoothness, making it challenging for many metrics. As shown in Figure 10, INRIQ provides much stronger sensitivity than PSNR, SSIM, LPIPS, MS-SSIM, and CLIP.

At the key transition (**Level 2→3**), INRIQ attains a sensitivity of **2.48**, while LPIPS reaches only 1.64 and PSNR/SSIM remain below 0.21. This indicates INRIQ's superior ability to capture the structural degradation induced by motion blur. For **Level 3→4**, INRIQ continues to dominate with **1.09**, clearly surpassing LPIPS (0.40) and PSNR (0.18). Even at the strongest distortions (**Level 4→5**), INRIQ remains higher at **0.67**, while LPIPS and SSIM saturate below 0.50. These results show that INRIQ robustly tracks both moderate and severe motion blur degradations, where pixel- and feature-based metrics underestimate perceptual loss.

### A.1.5 DARKEN

Darkening reduces luminance nonlinearly, compressing dynamic range while preserving extreme values, which often makes the perceptual impact less evident to pixel-based metrics. As illustrated

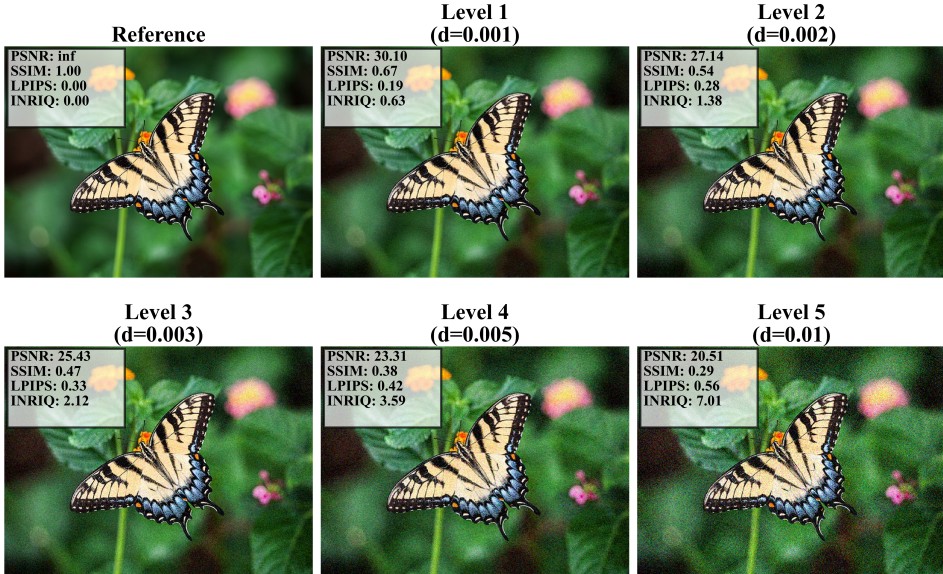

Figure 9: Visual progression of **White noise** across five severity levels. INRIQ exhibits substantially higher sensitivity, particularly in the early transition (Level 1→2), where traditional metrics underestimate the perceptual impact.

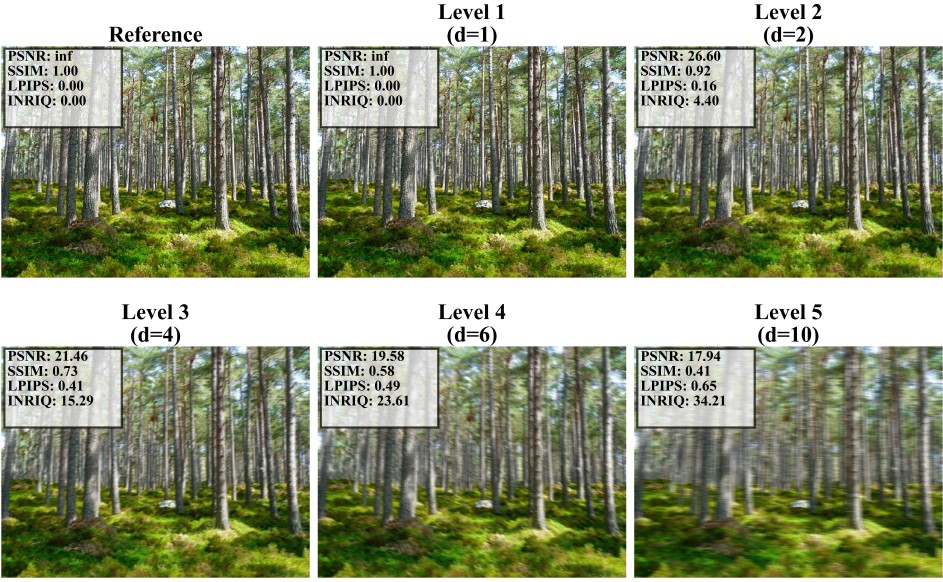

Figure 10: Visual progression of **Motion blur** across five severity levels. INRIQ demonstrates substantially higher sensitivity, especially in the mid-level transition (Level 2→3), where conventional metrics largely underestimate perceptual changes.

in Figure 11, INRIQ shows markedly higher sensitivity compared to PSNR, SSIM, LPIPS, MS-SSIM, and CLIP across all severity transitions.

At the earliest stage (**Level 1→2**), INRIQ records a sensitivity of **3.55**, while LPIPS remains at 2.36 and all others fall below 0.20. This trend continues at **Level 2→3** (**2.98** vs. LPIPS 2.46, PSNR 0.21, SSIM 0.03), highlighting INRIQ's ability to capture early luminance degradation more effectively than baselines. In the stronger transitions (**Level 3→4** and **Level 4→5**), INRIQ still dominates with **2.98** and **2.67**, while LPIPS remains at 2.39 and 2.15, and all other metrics stay well below 1.0. Overall, INRIQ robustly quantifies both subtle and severe brightness changes, where conventional metrics underestimate perceptual sensitivity.

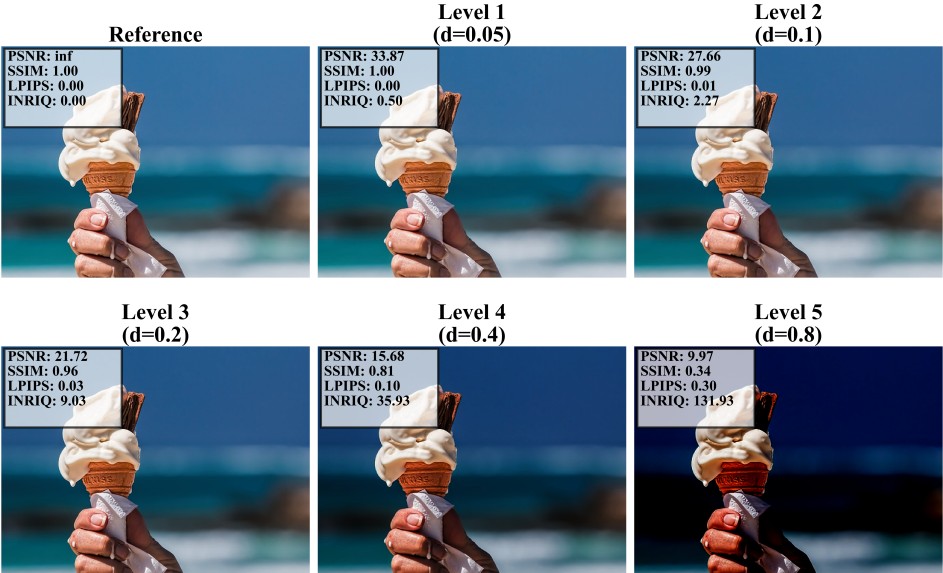

Figure 11: Visual progression of **Darken** across five severity levels. INRIQ exhibits consistently stronger sensitivity than all baseline metrics, particularly in the early and mid-level transitions, capturing perceptual luminance degradations more faithfully.

### A.2    BEYOND HUMAN PERCEPTION: WHY INRIQ APPEARS HIGHER

An important observation in our analysis is that INRIQ does not always correlate perfectly with human perception. This discrepancy becomes most evident in early transitions such as **White Noise (Distortion 11, Level 1→2)**, where INRIQ records a sensitivity of **1.18**, while LPIPS, PSNR, and SSIM remain at 0.49, 0.10, and 0.19, respectively. Similarly, in the case of **Impulse Noise (Distortion 13, Level 1→2)**, INRIQ attains **1.58**, far exceeding LPIPS (0.57) and PSNR/SSIM values below 0.05. Although the visual difference between the reference and degraded images in these transitions may appear minor to human observers, INRIQ identifies them as substantial degradations.

This behavior arises from INRIQ's construction: the score is derived from an implicit neural representation, where the entire signal is encoded in a continuous weight space. In this space, degradations are reflected not only in localized pixel differences but also in how the network parameters must adjust to preserve the global structure of the signal. As a result, INRIQ can register higher sensitivity scores than human observers might expect, effectively going *beyond human perception* by capturing perturbations that lie below subjective thresholds. .

The sensitivity analysis across the Kadid10k dataset further confirms this property. Compared against PSNR, SSIM, LPIPS, MS-SSIM, and CLIP, INRIQ consistently emerges as the most responsive measure.

## B    UTILIZATION OF LARGE LANGUAGE MODELS

During preparation of the manuscript, LLMs were used for polishing the writing style, correcting grammar, and rephrasing.

