# OpenReview forum: "INRIQ: Implicit Neural Representation for Image Quality Assessment"
_ICLR.cc/2026/Conference — ICLR 2026 Conference Withdrawn Submission_

### Official Review · Reviewer_QmLS · 2025-10-24

**Soundness:** 2
**Presentation:** 3
**Contribution:** 2
**Rating:** 4
**Confidence:** 4

**Summary:**

The paper proposes INRIQ, a full-reference IQA approach that reframes similarity measurement as a parameter-space discrepancy between implicit neural representations (INRs). Concretely, the authors (i) overfit an INR to the clean reference image, (ii) expose the same network to the degraded image and compute per-pixel loss gradients with respect to the INR parameters, (iii) aggregate these into a diagonal Fisher information vector, and (iv) compute a symmetrized KL divergence between the Fisher diagonals of reference and degraded images as the final score. The method is optionally bootstrapped with meta-learned INR initializations trained on Vimeo90k to accelerate fitting. The authors introduce a new evaluation metric called relative sensitivity to distortion severity. Evaluation focuses on KADID-10k where INRIQ often exceeds some other IQA metrics.

**Strengths:**

1. Conceptualy novel idea in the field of IQA metrics. The authors use representation sensitivity in INR weight space instead of pixel/feature-space metrics. The proposed symmetrized KL of Fisher diagonals represents a symmetric, non-negative divergence with a relatively intuitive interpretation.
2. Sensitivity analysis. The paper explicitly targets and quantifies responsiveness to incremental degradation, where traditional metrics can saturate; INRIQ shows strong gains on blur, compression, brightness/contrast, and several noise types.
3. Breadth beyond natural images. A small MRI undersampling demo suggests the framework may transfer to other imaging regimes.

**Weaknesses:**

1. The paper does not report standard IQA correlations (SRCC/PLCC/KRCC) against human judgments. The sole primary criterion is relative sensitivity, which is not a standard acceptance criterion for IQA and does not validate perceptual fidelity. Moreover, the authors explicitly acknowledge that INRIQ may depart from human perception (over-sensitive to low-level noise at small severities). This raises a serious concern about suitability as a general IQA measure.
2. The authors compare INRIQ with a limited number of IQA metrics. Recent strong FR metrics such as DreamSim, TOPIQ, and others are absent from the main quantitative analysis. This weakens claims such as being “the most sensitive similarity measure”. Additionally, for evaluation only KADID-10k was used. The common practice for IQA paper is to use several different datasets (LIVE, CSIQ, PIPAL, etc.)
3. The method requires overfitting an INR for each reference image and then computing per-pixel gradients for each degraded image. This pipeline raises a concern about time complexity and comparison with other methods, which do not require overfitting.

**Questions:**

1. The primary evaluation metric in the paper is relative sensitivity, which is not a standard acceptance criterion for IQA. Is relative sensitivity a better choice for evaluating IQA metrics? If yes, why? Could you provide correlation comparisons with other IQA metrics?
2. What is the time complexity of INRIQ?

---

### Official Review · Reviewer_bXfj · 2025-10-31

**Soundness:** 3
**Presentation:** 2
**Contribution:** 3
**Rating:** 4
**Confidence:** 4

**Summary:**

The authors propose INRIQ, a metric derived from the Fisher information of an INR fitted to a reference image. By comparing how much the INR’s parameters must change to reproduce a degraded image, INRIQ quantifies perceptual similarity in parameter space. The symmetric KL divergence between Fisher diagonals serves as the similarity measure. Experiments on KADID-10k show that INRIQ exhibits higher “relative sensitivity” than PSNR, SSIM, LPIPS, MS-SSIM, and CLIP across multiple distortion types.

**Strengths:**

1) The paper introduces a fresh way to think about perceptual similarity—through the geometry of INR parameter spaces rather than conventional image or feature spaces.

2) The Fisher-based framework provides a degree of mathematical interpretability, linking sensitivity analysis with perceptual response.

3) Visual comparisons (e.g., JPEG2000, color diffusion, MRI undersampling) nicely illustrate the metric’s responsiveness to subtle degradations.

**Weaknesses:**

- No correlation analysis (SRCC, PLCC, RMSE) with human Mean Opinion Scores (MOS) is presented, even though it is standard in IQA.
- The evaluation is restricted to KADID-10k, which is primarily synthetic and does not fully test generalization to real-world degradations (e.g., LIVE, TID2013, PIPAL, SPAQ).
- The MRI example is interesting but anecdotal—no quantitative evaluation or domain-specific baselines are given.
- It is unclear why Fisher diagonals capture perceptual change better than direct weight or activation differences, beyond intuition.

**Questions:**

- The meta-learning stage on Vimeo90k is mentioned but not quantified—what improvement does it bring versus random initialization?
- How does INRIQ behave when the reference image itself is noisy or of low quality? Is it robust to imperfect references?
- Have you tested cross-dataset generalization, e.g., training on KADID-10k and evaluating on PIPAL or LIVE?

---

### Official Review · Reviewer_efmD · 2025-11-01

**Soundness:** 2
**Presentation:** 2
**Contribution:** 2
**Rating:** 2
**Confidence:** 4

**Summary:**

The paper proposes INRIQ, a full-reference image quality assessment (IQA) metric that reframes similarity as the extent to which an Implicit Neural Representation (INR) would need to adapt to a degraded image to match the reference image. It computes per-pixel gradients on a frozen, reference-trained INR, aggregates them into a diagonal Fisher information vector, and measures discrepancy via a symmetrized KL between the reference and degraded Fisher diagonals. The initialization is a meta-learning approach on Vimeo90k to enable faster adaptation. Experiments on KADID-10k report higher relative sensitivity to increasing distortion severities than PSNR/SSIM/MS-SSIM/LPIPS/CLIP, with qualitative examples including JPEG2000 and MRI undersampling.

**Strengths:**

The work offers a proposal for a full-reference IQA by shifting comparison from pixel/feature space to the adaptation dynamics of an INR fitted to the reference, instantiated with a  Fisher-diagonal plus symmetric-KL construction and a pragmatic meta-learned initialization that makes the idea usable. The methodological core is compact and implementable: it overfits INR, accumulates per-pixel gradients into Fisher statistics, compares via KL, and the empirical section systematically probes many distortion types with a consistent “relative sensitivity” lens across severities. The narrative is easy to follow, and the visuals help convey how responses evolve as degradations intensify, while the MRI undersampling case suggests the approach can be used in synthetic distortions. This proposal could be valuable in settings where human MOS labels are scarce, and it shows a connection between INR properties (e.g., spectral bias, parameter sensitivity) and perceived image quality, which could influence assessment, restoration, and compression research.

**Weaknesses:**

The main limitations are in the evidence and scope. The evaluation centers on a custom “relative sensitivity” metric and reports averages across KADID-10k, but does not include standard human-opinion correlations (SRCC/PLCC vs MOS/DMOS) on LIVE/TID2013/KADID-10k, so the key “perceptual” claim remains unverified against human judgments; moreover, the appendix explicitly notes cases where INRIQ reacts more strongly than humans at low distortion levels, underscoring a potential misalignment that should be quantified rather than asserted away. Methodologically, the comparison set is narrow: the main table covers PSNR/SSIM/MS-SSIM/LPIPS/CLIP but omits strong full-reference baselines such as DISTS, FSIM, VIF, and PieAPP that often set the bar in IQA. Adding these (with careful configuration) would make the ranking more credible.  The metric numerically uses only the diagonal Fisher with a symmetric KL and explicitly excludes degenerate cases when aggregating sensitivity. The paper should report how often such exclusions occur, adopt and consider low-rank/block-diagonal alternatives to test robustness to small entries. The empirical scope is narrow: apart from an illustrative MRI figure, experiments primarily involve synthetic distortions on a single dataset. Adding cross-dataset tests and stress tests for misalignment (small warps/crops), content-preserving photometric transforms, and AI-generated content would clarify robustness and generality.

**Questions:**

1. Report SRCC/PLCC vs MOS/DMOS on LIVE, TID2013, and KADID-10k with significance tests. Where INRIQ departs from human judgments (e.g., low severities), can a monotone calibration recover alignment without sacrificing sensitivity?

	2. Add strong FR IQA methods (DISTS, FSIM, VIF, PieAPP) and tuned LPIPS/CLIP variants under identical preprocessing/resolution. Provide per-distortion and pooled comparisons with statistical testing.

	3.  Give a precise mathematical definition, all hyperparameters, and aggregation rules across severities; report confidence intervals/effect sizes, check monotonicity, analyze dependence on severity granularity, and clarify handling of degenerate cases.

	4.  Provide clock time, GPU memory, and iteration counts per image at multiple resolutions; detail sampling, batch sizes, and gradient accumulation; analyze cost vs INR depth/width/activation, and give guidance for feasible deployment.

	5. Stress-test small misalignments/crops and photometric transforms; evaluate on AI-generated content; add cross-dataset transfer and at least one real-world pipeline (e.g., codec chain or medical modality) with task-relevant metrics.

---

### Official Review · Reviewer_rC68 · 2025-11-02

**Soundness:** 2
**Presentation:** 2
**Contribution:** 1
**Rating:** 2
**Confidence:** 4

**Summary:**

Implicit neural representations (INRs) of images are multilayer perceptrons (MLPs), with special activation functions, that learn the transform from the spatial coordinates into the image values. When one of these MLPs is trained to learn one image, its parameters contain the information in the image. In this work the authors propose to use the above fact (image information may be contained in the parameters of MLPs), to measure "subjective" differences between two images using a measure of the differences between the parameters corresponding to each image. They dont measure L2 differences nor other measures of alignment between parameters (such as cosine, Canonical Correlation, or Centered Kernel Alignment). Instead, they use the symmetric Jensen-Shannon divergence between diagonal approximations of the Fisher information matrices for the parameters of the original and the distorted images.

The proposed method is certainly a distance metric which is quite sensitive to image distortions, but in contrast with the goal stated in the introduction (predicting human-perceived distances), they give no concluding evidence of the correlation between their measure and subjective ratings of distortion. The authors show that their metric is more sensitive to distortions than other standard metrics, but if this increased sensitivity does not correlate with human opinion, what is this new metric serving for?

**Strengths:**

The idea of measuring differences between images using the differences between the parameters of a network that represents the images as opposed to measuring differences between the images in certain feature space is (to my knowledge) new.

**Weaknesses:**

(A) There is no evidence of prediction of human opinion.
Systematic experiments (table 1) only show that a definition of sensitivity (averaged over images and different distortion levels for each separate distortion) is bigger in their metric than in other metrics used to predict human opinion. This increased sensitivity does not mean that this new metric better predicts human opinion.
The other results shown (e.g. figs 1, 4-6 in the main text and figs. 7-11 in the appendix) only show that the proposed metric "does change" with distortions, but by no means show that this variation is more similar to human opinion of distortion than the variation displayed by other metrics. This is explicitly acknowledged in the first paragraph of Appendix A.2 (lines 849-854).

One example of the above: human sensitivity to white noise is known to be a saturating function of the standard deviation (contrast) of the noise [Daly90, Watson&Solomon97]. However, the values reported in Fig. 9 show that while SSIM and LPIPS saturate with \sigma of the noise, the proposed measure increases exponentially (contrary to human opinion).

Failure in giving a measure that displays correlation with human opinion is in clear contradiction with the goal expressed in the abstract "as the severity of degradation increases or decreases, existing similarity measures often fail to respond proportionally, revealing their limited sensitivity to the true perceptual progression of signal quality", and in the introduction, "we bridge the gap between principled signal processing and perceptually aligned evaluation".  Unless the authors do not provide evidences of correlation with human opinion, the paper should be completely reformulated as a "sensitive metric of a yet-to-know purpose", as opposed to a "metric to predict human-perceived distances".

(B) The authors convey an arguable (non-human) intuition for the metrics.
They suggest that a desirable property in a metric is an increased sensitivity at higher distortion levels (e.g. in lines 300-301 and 333-335), and this is not how humans perceive distortions: in fact, human sensitivity saturates and decreases for larger amounts of distortion [Daly90, Watson&Solomon97].

(C) The authors do not explain why their proposal could work.
For instance, as one reads through the work this idea (and problem) appears seral times:
- In lines 96-98 the authors say: "bridge pixel-level fidelity and perceptual alignment in a unified framework, while retaining interpretability".
Do they mean that differences in weights of MLPs are (qualitatively) interpretable?
They dont even describe what is the MLP architecture they use.
- In lines 123-125 the authors say: "this reframes similarity evaluation as a problem of analyzing the sensitivity dynamics of a learned representation, which naturally aligns with perceptual distinctions that conventional metrics fail to
capture".
Why "naturally"?. It is clear that there will be a relation, but this relation is not necessarily linear nor easy to understand (as pointed out by the fact that, in the end there is not a linear correlation with human opinion).
- In lines 269-271 the authors say:
"If the degraded image is perceptually close to the reference, then its Fisher diagonal will closely match F_ref, resulting in a small KL divergence. If distortions significantly alter the structure of the signal, the induced Fisher diagonal will differ, and the KL divergence will be large."
Yes, but why (linearly) proportional to perceived distortion?. Why other differences between the coefficients do not work? (as acknowledged in lines 207-215).

REFS:

[Daly90] Daly S. Application of a noise-adaptive Contrast Sensitivity Function to image data compression. Optical Engineering 29 (1990) 977–987

[Watson&Solomon97] Watson A, Solomon J. A model of visual contrast gain control and pattern masking. JOSA A 14 (1997) 2379–2391

**Questions:**

(A) Please include a table with Pearson correlation between your distance and the Mean Opinion Score of different subjectively rated databases such as TID2008, TID2013, and KADID. If the results are not satisfactory, the work could not be accepted in its current form: a different justification for the work (other than reproducing human opinion) would be necessary.

(B) Please remove unsupported intuitions about how the sensitivity of a perceptual metric should be (e.g. increased sensitivity for larger distortions).

(C) Some intuition should be given about why measuring differences between parameters in this specific way is better than other options.

---

### Note · Authors · 2025-11-13

I have read and agree with the venue's withdrawal policy on behalf of myself and my co-authors.